# Macrophages and Urokinase Plasminogen Activator Receptor System in Multiple Myeloma: Case Series and Literature Review

**DOI:** 10.3390/ijms241310519

**Published:** 2023-06-23

**Authors:** Paola Manzo, Valentina Giudice, Filomena Napolitano, Danilo De Novellis, Bianca Serio, Paolo Moscato, Nunzia Montuori, Carmine Selleri

**Affiliations:** 1Hematology and Transplant Center, University Hospital “San Giovanni di Dio e Ruggi d’Aragona”, 84131 Salerno, Italy; pamanzo@unisa.it (P.M.); vgiudice@unisa.it (V.G.); danilo.denovellis@sangiovannieruggi.it (D.D.N.); bianca.serio@sangiovannieruggi.it (B.S.); 2Department of Medicine and Surgery, University of Salerno, 84081 Baronissi, Italy; 3Department of Translational Medical Sciences, University of Naples “Federico II”, 80138 Naples, Italy; filomena-napolitano88@hotmail.it (F.N.); nunzia.montuori@unina.it (N.M.); 4Rheumatology Unit, University Hospital “San Giovanni di Dio e Ruggi d’Aragona”, 84131 Salerno, Italy; paolo.moscato@sangiovannieruggi.it

**Keywords:** macrophages, urokinase plasminogen activator receptor, multiple myeloma

## Abstract

The microenvironment plays an essential role in multiple myeloma (MM) development, progression, cell proliferation, survival, immunological escape, and drug resistance. Mesenchymal stromal cells and macrophages release tolerogenic cytokines and favor anti-apoptotic signaling pathway activation, while the urokinase plasminogen activator receptor (uPAR) system contributes to migration through an extracellular matrix. Here, we first summarized the role of macrophages and the uPAR system in MM pathogenesis, and then we reported the potential therapeutic effects of uPAR inhibitors in a case series of primary MM-derived adherent cells. Our preliminary results showed that after uPAR inhibitor treatments, interleukein-6 (mean ± SD, 8734.95 ± 4169.2 pg/mL vs. 359.26 ± 393.8 pg/mL, pre- vs. post-treatment; *p* = 0.0012) and DKK-1 levels (mean ± SD, 7005.41 ± 6393.4 pg/mL vs. 61.74 ± 55.2 pg/mL, pre- vs. post-treatment; *p* = 0.0043) in culture medium were almost completely abolished, supporting further investigation of uPAR blockade as a therapeutic strategy for MM treatment. Therefore, uPAR inhibitors could exert both anti-inflammatory and pro-immunosurveillance activity. However, our preliminary results need further validation in additional in vitro and in vivo studies.

## 1. Introduction

Multiple myeloma (MM), a malignant hematological disorder characterized by the clonal expansion of malignant plasma cells in the BM, is still considered an incurable disease because of the high incidence of relapse/refractoriness even to novel targeted therapies [1]. In MM, malignant cell growth and survival strongly rely on microenvironment stimuli, as neoplastic plasma cells are unlikely to survive outside the bone marrow (BM) [2,3,4,5,6], and complex cross-talks between stromal cells and neoplastic clones might favor the transition from premalignant conditions to active MM [6,7,8,9]. The microenvironment plays an important role in modulating immune surveillance against tumor cells and in facilitating tumor growth and angiogenesis [10]. Within the BM microenvironment in MM, mesenchymal stem cells (MSCs) and macrophages are essential for plasma cell proliferation, angiogenesis, immunotolerance, and drug resistance [11,12,13,14,15]. Indeed, CD68^+^ and CD163^+^ MM-associated macrophages at diagnosis are proposed as prognostic factors [16,17].

Macrophages, phagocytic cells, play an important role in innate immune responses with pleiotropic actions [4] and are divided into two phenotypically and functionally different subpopulations: anti-inflammatory M1 and pro-inflammatory M2 cells [18]. Classically activated M1 macrophages are antigen-presenting cells, have phagocytic activity, are positive for CD80, CD86, CD16, and CD32, and their polarization is induced by proinflammatory T helper (Th)1 cytokines (e.g., interferon[IFN]-γ) [4,18]. Conversely, M2 macrophages have anti-inflammatory properties, appear later for the resolution of inflammation, are induced by Th2 cytokines (mainly IL-4 and IL-13), and are positive for CD206 [4,9,18,19,20]. Impaired M1–M2 switch during inflammation or altered polarization can promote cancer initiation and progression [4].

In newly diagnosed MM, the BM is populated by CD68^+^CD163^+^ macrophages that support the survival, proliferation, and drug resistance of MM cells [7,9,12,16,17]. Circulating monocytes accumulate in the BM because neoplastic plasma cells produce monocyte-attracting chemokines, such as chemokine ligand 2 (CCL2), and release soluble cleaved uPAR (c-suPAR) [6,11,12,21,22]. Once in the BM, macrophages initially polarize toward the M1 phenotype and start sustaining the neoangiogenesis, immune escape, and drug resistance of neoplastic plasma cells [12,23,24,25]. Subsequently, M2 macrophages support the survival and proliferation of MM cells through the autocrine secretion of IL-6 [7,24,25,26]. Indeed, increased BM frequency of M2 macrophages is associated with poor prognosis and advanced disease, while no expansion is observed in premalignant conditions [7,16,19,27,28,29,30]. Moreover, M2 macrophages accumulate in the necrotic center of plasmacytomas where neoangiogenesis and tissue remodeling are required for tumor growth, while pro-inflammatory M1 macrophages are distributed in the invasive border to continuously recruit immune cells [12]. M2-polarized macrophages are also unable to phagocyte or direct the adaptive response toward neoplastic plasma cells, likely because of the CD47 inhibitory signal of macrophage phagocytosis by MM cells [12,31]. In in vitro co-culture systems with MSCs and MM cells, macrophages demonstrate an increased expression of several cytokines and vascular endothelial growth factors, favoring MM cell proliferation and neoangiogenesis [4,7,9,12,14,32,33]. MM-associated macrophages also release urokinase-type plasminogen activator (uPA), matrix metalloproteinases (MMPs), and macrophage-colony stimulating factor (M-CSF) that contribute to osteoclast differentiation and bone remodeling [34]. Furthermore, in in vitro co-culture systems with macrophages, MM cells acquire resistance to various anti-myeloma agents, such as melphalan and bortezomib [4,35]. Indeed, macrophages prevent drug-induced apoptosis by reducing caspase-3 cleavage through selectin glycoprotein ligand 1 (PSGL-1) and intercellular adhesion molecule-1 (ICAM-1) on MM cells and E/*p* selectins and CD18 on macrophages [12,36].

uPAR, a glycosyl-phosphatidylinositol (GPI)-anchored surface receptor, binds its ligand uPA and localizes its proteolytic activity on cell membranes [37,38]. In physiological conditions, uPA and uPAR are expressed by leukocytes, endothelial cells, and fibroblasts, and regulate cell adhesion and migration [39,40,41]. Plasmin-mediated matrix proteolysis induces the release of several growth factors via the extracellular matrix, such as VEGF [42]. uPAR also regulates cell adhesion through full-length uPAR-integrin binding, especially fibronectin α3β1-α5β1 receptors and macrophage 1 antigen (Mac-1) [43]. Monocytes express uPAR, especially at the migration front providing directionality and circumscription to proteolytic matrix remodeling [44,45]. Monocyte chemotaxis is also guided by N-formylated Met-Leu-Phe (fMLF) peptides recognized by fMLF receptors FPR1, FPR2, and FPR3 [46,47]. Once uPAR is cleaved, an SRSRY sequence is exposed and can bind and activate FPR2, acting as a potent chemoattractant [46]. uPAR also influences monocyte-to-macrophage differentiation through Mac-1 (CD11b/CD18) signaling, while PAI-1 inhibits both differentiation and adhesion [48,49,50]. Moreover, Mac-1 interacts with uPAR during phagocytosis, leading to RAC-1 activation, a key regulator of phagocytosis [51]. Moreover, uPAR^+^ tumor cells could modulate macrophage polarization toward the M2 phenotype through TGF-β signaling pathways after uPA-uPAR cleavage-dependent activation [19,52,53], while TGF-β induces uPA expression on cancer cells in positive feedback [52,53,54,55,56]. Therefore, targeting uPAR could prevent monocyte recruitment and reduce malignant plasma cell survival, diffusion, and osteolytic lesions; however, blocking it may affect macrophage recruitment, adhesion, differentiation, phagocytosis, and polarization. The prevention of uPA binding to its uPAR receptor only results in modest effects on tumor progression and metastatic invasion [57]. Therefore, novel therapies are targeting other uPAR ligands, including integrins and vitronectin [57,58].

## 2. Preliminary Study: Potential Clinical Relevance of uPAR Blockade in MM

In this work, we reported potential therapeutic strategies to target macrophages and microenvironments by testing several uPAR inhibitors in a case series of MM-derived adherent primary cell lines. First, adherent cells from healthy and MM subjects were characterized by morphology and flow cytometry immunophenotyping. At light microscopy examination, adherent cells had fibroblast-like spindle shapes; however, diffuse monocyte-like cells were also observed (Figure 1).

The presence of these two different populations was also confirmed by flow cytometry immunophenotyping showing a mesenchymal-like population with positivity for CD90 and negativity for CD14, CD45, and HLA-DR [59,60], while a small fraction (0.2–2.6% of total nucleated cells) had an M2-macrophage-like phenotype with positivity for CD163 and CD206 (Figure 2).

IL-1, TNF-α, IL-15, IL-10, MIP-1α, and 1β were not measurable at baseline or after treatment. Conversely, IL-6 (mean ± SD, 7897.99 ± 4255.5 pg/mL vs. 962.2 ± 1462.5 pg/mL, MM vs. controls; *p* = 0.0013 by uncorrected Fisher’s LSD test) and DKK-1 (mean ± SD, 6516.44 ± 5842.6 pg/mL vs. 1041.1 ± 652.1 pg/mL, MM vs. controls; *p* = 0.0073 by uncorrected Fisher’s LSD test) were significantly increased in the media MM patients after 21 days of culture compared to healthy controls (Figure 3). HGF was slightly increased in MM, although not significantly (mean ± SD, 987.16 ± 1323.2 pg/mL vs. 109.1 ± 57.2 pg/mL, MM vs. controls; *p* = 0.6258 by uncorrected Fisher’s LSD test).

Next, MM-derived mesenchymal cells were treated with C6 or C37 for 1 h, and cytokine levels were measured in culture media (Figure 4). All four compounds induced a significant decrease in IL-6 and DKK-1 levels (all *p* < 0.05 by uncorrected Fisher’s LSD test), more markedly after C6 treatment. Indeed, IL-6 levels significantly decreased to 522.6 ± 265.9 pg/mL with C37, and to 79.25 ± 10.4 pg/mL with C6. Similarly, DKK-1 significantly decreased to 60.95 ± 86.2 pg/mL with C37, and to 6.95 ± 9.8 pg/mL with C6.

After C6 treatment, IL-6 (mean ± SD, 8018.35 ± 4121.5 pg/mL vs. 324.59 + 362.4 pg/mL, pre- vs. post-treatment; *p* = 0.0003) and DKK-1 levels (mean ± SD, 7199.65 ± 5738.2 pg/mL vs. 56.1 ± 51.3 pg/mL, pre- vs. post-treatment; *p* = 0.0007) were almost completely abolished (Figure 5).

## 3. Discussion

In MM, uPAR has been proposed as a prognostic marker of disease progression, as its levels correlate with tumor burden and increased uPAR^+^ plasma cells can identify MM progression early on [22,39,61]. Immature CD45^+^ MM cells show higher uPAR levels compared to mature CD45^−/dim^ plasma cells [62]; however, the exact mechanisms triggered by the uPA-uPAR system for promoting MM cell proliferation are still unclear. In a preliminary study, we tested two uPAR inhibitors, C6, and C37, identified by the structure-based virtual screening of the National Cancer Institute (National Cancer Institute, National Health Institutes, Bethesda, MD, USA) in primary MM cells [63]. These molecules have shown selective activities in cell adhesion to vitronectin inhibition and in uPAR interaction with the fMLF family of chemotaxis receptors (fMLF-Rs) [63]. C37 is also under investigation for the development of novel antifibrotic therapies, because of its ability to reduce oxidative stress in dermal fibroblasts isolated from the skin of patients with systemic sclerosis [64]. These compounds have demonstrated great ability in vitro to reduce extracellular matrix invasion, thus representing new promising therapeutic targets. In this work, we reported the potential therapeutic effects of C6 and C37 uPAR inhibitors in a case series of MM-derived adherent primary cell lines.

Neoplastic plasma cells produce HGF that is cleaved and activated by uPA and plasmin, leading to IL-11 secretion by osteoblasts and tumor cell invasion mediated by osteoclasts, as uPA inhibition also halts MM invasion [39,65,66]. MM plasma cells also interact with vitronectin and fibronectin through an α5β3 integrin and adhere to the extracellular matrix, increasing cell proliferation and MMPs and uPA secretion [67]. However, our preliminary data showed that no significant variations were observed in the culture media of MM primary cell lines at baseline and after uPAR inhibitor treatments, suggesting that HGF-related pathways are not activated outside the BM niche without extracellular matrix interaction.

MM-associated macrophages actively contribute to immunosuppression, disease progression, and drug resistance; therefore, this cell population is emerging as a potential therapeutic target [9,31,68,69]. Numerous strategies have been evaluated to promote the anti-tumor potential of macrophages, such as the use of JAK1/2 inhibitor, ruxolitinib, to reverse M2 polarization in favor of the M1 phenotype in in vitro and in vivo MM models [30]. Moreover, macrophage genetic reprogramming can be performed at the epigenetic level using histone deacetylase inhibitors (HDACi), TLRs agonists, or cytokine modulators, such as with a combination of GM-CSF (a pro-M1 molecule) and 4-iodo-6-phenyl-pyrimidine (an inhibitor of a pro-M2 cytokine, MIF) [31,70,71]. Our data show stronger anti-inflammatory and pro-immunosurveillance activities in MM primary cells of C6 compared to C37, likely because of its structure and binding modes to uPAR. In particular, C6 mimics vitronectin residues responsible for insertion into uPAR’s surface, making H-bonding and hydrophobic interactions; at the same time, C6 interacts with R91 and S88, two key residues of the SRSRY sequence, that could interfere with uPAR/fMLF-R crosstalk. Conversely, C37 only partially contacts this sequence [63].

The microenvironment plays an essential role in cancer development and progression, as also described in MM. Indeed, MM cell proliferation, survival, immunological escape, and drug resistance are deeply influenced by tumor-associated cell populations, including mesenchymal stromal cells and macrophages [72]. These cells alter the tumor microenvironment with the release of tolerogenic cytokines, while also favoring anti-apoptotic signaling pathway activation and drug resistance by cell-to-cell contacts with neoplastic clones. The uPA/uPAR system plays an important role in cancer development and progression, especially in favoring migration through the extracellular matrix; however, several other mechanisms might be involved in MM progression, thus proposing the uPA/uPAR system as a candidate therapeutic target [63,64,73]. Our preliminary results showed that uPAR inhibition exerted a potent anti-inflammatory effect by almost abolishing IL-6 and DKK-1 secretion; in particular, DKK-1 glycoprotein, an inhibitory component of the Wnt pathway, promotes pathological type 2 cell-mediated inflammation and immune evasion and induces immunosuppressive macrophages in several cancers [74,75]. In MM, DKK-1 is highly expressed by myeloma cells and regulates myeloma bone disease, defined as the displacement of hematopoiesis and the formation of osteolytic lesions via osteoblast inhibition and osteoclast activation [76]. Moreover, DKK-1 is highly secreted in bortezomib-resistant MM by sustaining drug resistance through CD138 downregulation and CKAP4 receptor-mediated NF-κB activation [77]. In addition, IL-6 contributes to DKK-1-mediated drug resistance, as this cytokine stimulates CKAP4 expression [77]. Therefore, the simultaneous abolishment of IL-6 and DKK-1 by C6 and C37 uPAR inhibitors in our encouraging preliminary results proposes these molecules as promising targeted therapy in MM and opens new perspectives on the pathogenic role of uPAR in MM and drug resistance, as uPAR inhibitors could exert both anti-inflammatory and pro-immunosurveillance activity. However, our preliminary results need further validation in additional in vitro and in vivo studies.

## 4. Materials and Methods

### 4.1. BM-MSC Culture Conditions

To preliminarily investigate the potential anti-cancer effects of C6 and C37 in MM, six MM patients (M/F, 4/2; mean age, 59 years old; range, 43–68 years) were included in this study, and BM specimens were collected in ethylenediaminetetraacetic acid (EDTA) tubes at diagnosis according to the Declaration of Helsinki and protocols approved by the local ethic committee (Campania Sud; prot./SCCE n. 24988). BM was obtained from four healthy controls (M/F, 3/1; mean age, 39 years old; range, 22–55 years old). Briefly, total BM aspirate was seeded at a concentration of 50,000 total nucleated cells/cm^2^ in T75 plastic flasks in minimum essential medium alpha (α-MEM) supplemented with 1% Glutagro^TM^ (Corning, Manassas, VA, USA), 10% fetal bovine serum (FBS) and 1% penicillin/streptomycin (Pen/Strep), and were incubated at 37 °C, 5% CO_2_, and 95% relative humidity, as previously described [78,79]. After 72 h, nonadherent cells were removed, while adherent cells were kept in culture until day 21. Once cell cultures reached 70–80% confluence, cells were detached using 0.05% trypsin-0.53 mM EDTA and washed with 1× of phosphate-buffered saline (PBS) (Corning Cellgro, Manassas, VA, USA).

The mesenchymal phenotype was confirmed according to the International Society of Cellular Therapy guidelines: (i) the ability to adhere to tissue culture plastics; (ii) a fibroblast-like spindle shape; and (iii) a characteristic immunophenotype by flow cytometry [59].

### 4.2. Flow Cytometry

For immunophenotyping, a minimum of 1 × 10^5^ cells were stained with the following antibodies: 2.5 μL of fluorescein isothiocyanate (FITC)-conjugated anti-CD90 or 5 μL of FITC-conjugated anti-HLA-DR; 5 μL of allophycocyanin (APC)-conjugated anti-CD14 or CD163; and 10 μL of phycoerythrin cyanin 7 (PC7)-conjugated anti-CD45 or anti-CD206 (all antibodies from Beckman Coulter, Fullerton, CA, United States). Cells were incubated at room temperature (RT) for 20 min in the dark, washed with phosphate-buffered saline (PBS, Gibco™, Grand Island, NY, USA), and resuspended in 300 μL of the same buffer for acquisition, performed on a BD FACSVerse flow cytometer (Becton Dickinson, Franklin Lakes, NJ, USA) equipped with blue (488 nm) and red lasers (628 nm). PMT voltage setting and compensation were carried out using single-color controls for each fluorochrome and an unstained sample as a negative control. All samples were run with the same PMT voltages, and a minimum of 30,000 events were recorded. Kaluza software (v.2.1, Beckman Coulter) was used for post-acquisition compensation and analysis, as previously described [78,79]. Cells were first identified using linear parameters (forward scatter area (FSC-A) vs. side scatter area (SSC-A)), and double cells were excluded (area vs. height, FSC-A vs. FSC-H) (Figure 2A). On single cells, CD90, CD45, CD14, and HLA-DR expression was investigated, while on CD163^+^CD90^-^ cells, CD206 expression was further studied (Figure 2B).

### 4.3. Anti-uPAR Treatment and Cytokine Detection

Next, a minimum of 4 × 10^3^ cells were treated with C6 or C37 at a final concentration of 50 µM for 1 h at 37 °C, 5% CO_2_, and 95% relative humidity. The culture medium was collected on day 21 and after uPAR inhibitor treatment. For cytokine detection in culture medium, a bead-based multiplex custom immunoassay (9-plex LEGENDplex^TM^ Custom Panel; BioLegend, San Diego, CA, USA) was used for the measurement of IL-1, IL-6, TNF-α, HGF, IL-15, IL-10, macrophage inflammatory protein (MIP)-1α and 1β, and Dickkopf-related protein 1 (DKK1). Samples were diluted 1:50 with fresh complete α-MEM and processed according to the manufacturers’ instructions. Samples were run in duplicate, and cytokine quantification was performed by converting median fluorescence intensity to concentrations using a calibration made with standards as per the manufacturer’s instructions. A total of 2400 total beads (1200 Beads A and 1200 Beads B) was recorded using a BD FACSVerse cytometer (BD Biosciences, Franklin Lakes, NJ, USA), and post-acquisition analysis was carried out using LEGENDplex™ Data Analysis Software Suite (BioLegend, San Diego, CA, USA).

### 4.4. Statistical Analysis

Data were collected in a spreadsheet and analyzed using Prism (v.9.5.1; GraphPad software, San Diego, CA, USA). A two-group comparison was carried out via an unpaired t-test, while a multiple-group comparison was carried out via ANOVA with an uncorrected Fisher’s LSD test. A *p* value < 0.05 was considered statistically significant.

## Figures and Tables

**Figure 1 ijms-24-10519-f001:**
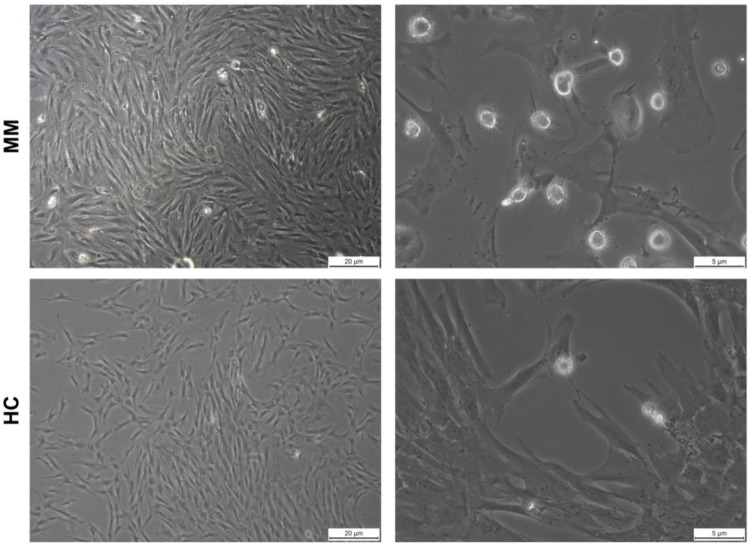
Brightfield images of adherent cells. Images were captured using a Leica DMIL LED microscope and acquired by Leica DFC425 C Camera after 14 days of culture of multiple myeloma (MM)-derived adherent cells (**upper panels**) or healthy-control (HC)-derived cells (**bottom panels**).

**Figure 2 ijms-24-10519-f002:**
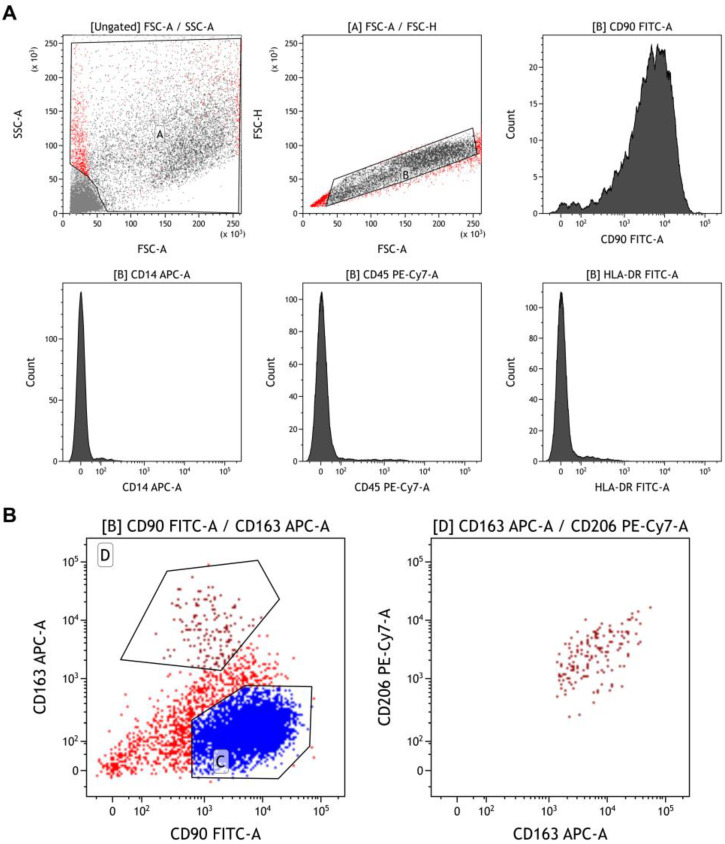
Immunophenotyping of adherent MM-derived cells. (**A**) Expression of CD90 (a mesenchymal marker), CD14, CD45, and HLA-DR was investigated on single cells gated first on linear parameters (FSC-A vs. SSC-A; gate A), and then by removing double cells (FSC-A vs. FSC-H; gate B). (**B**) Non-mesenchymal CD90^+^ cells were gated for CD163 expression (gate D) and further investigated for CD206. Mesenchymal CD90^+^ cells (gate C) were also identified.

**Figure 3 ijms-24-10519-f003:**
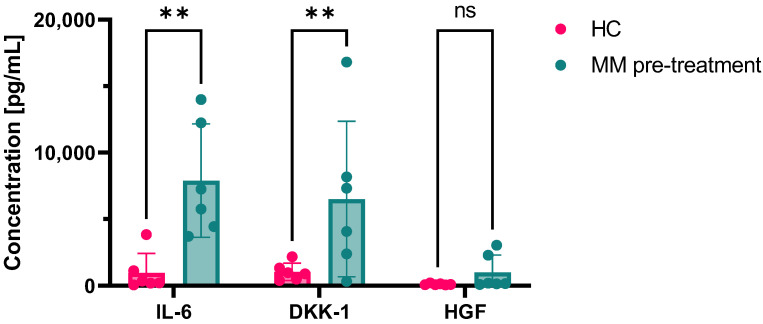
Cytokine production at baseline. Levels of interleukin-6 (IL-6), Dickkopf-related protein 1 (DKK1), and hepatocyte growth factor (HGF) in culture media were measured at baseline in healthy controls (HC) and multiple myeloma (MM) patients after 21 days of culture. ** *p* < 0.01; ns, not statistically significant.

**Figure 4 ijms-24-10519-f004:**
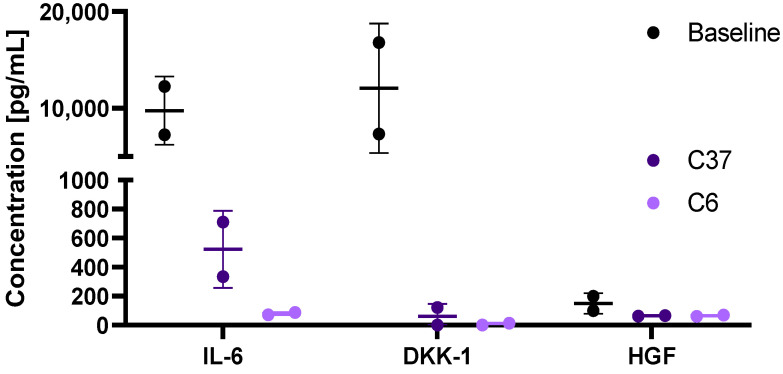
Cytokine levels after uPAR inhibitor treatment. Levels of interleukin-6 (IL-6), Dickkopf-related protein 1 (DKK1), and hepatocyte growth factor (HGF) in culture media were measured at baseline of multiple myeloma (MM) patients after 21 days of culture, and after treatment with uPAR inhibitors, C37 and C6.

**Figure 5 ijms-24-10519-f005:**
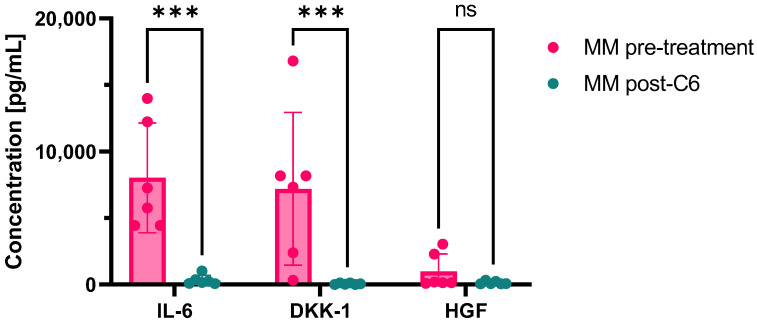
Cytokine levels after C6 treatment. Levels of interleukin-6 (IL-6), Dickkopf-related protein 1 (DKK1), and hepatocyte growth factor (HGF) in culture media were measured at baseline of multiple myeloma (MM) patients after 21 days of culture and after treatment with C6. *** *p* < 0.001; ns, not statistically significant.

## Data Availability

Data are available upon request by the authors.

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
