# Peer review of "Macrophages and Urokinase Plasminogen Activator Receptor System in Multiple Myeloma: Case Series and Literature Review"

_ijms, 2023, doi:10.3390/ijms241310519_

Round 1

Reviewer 1 Report

The work of Dr, Manzo et al. shows the rationale to use uPAR inhibitors in MM.

Although this article is potentially interesting, it is poorly readable and should be reorganized.

Major Comments:

A more focused and well-organized introduction will improve the readability of the manuscript. More information about the importance to target uPAR should be provided into the introduction.

Most of text contained into the section in results section should be moved into material and methods section, as well as most of the text in material and methods section (line 279-294) should be deleted.

Minor changes are required

Author Response

Reviewer #1

The work of Dr, Manzo et al. shows the rationale to use uPAR inhibitors in MM. Although this article is potentially interesting, it is poorly readable and should be reorganized.

Major Comments:

Comment 1. A more focused and well-organized introduction will improve the readability of the manuscript. More information about the importance to target uPAR should be provided into the introduction.

Response to Comment 1. We thank the Reviewer for this comment, and we have completely reorganized the manuscript according to a Hypothesis paper for International Journal of Molecular Sciences journal’s format. We have shortened the introduction and added information on uPAR targeting.

Comment 2. Most of text contained into the section in results section should be moved into material and methods section, as well as most of the text in material and methods section (line 279-294) should be deleted.

Response to Comment 2. We thank the Reviewer for this comment, and we have completely rearranged the Materials and Methods section by also including separate paragraphs (4.1 BM-MSC culture conditions; 4.2 Flow cytometry; 4.3 Anti-uPAR treatment and cytokine detection; 4.4 Statistical analysis).

Reviewer 2 Report

Dear Authors:

Report on the review article entitled “Macrophages and urokinase plasminogen activator receptor system in multiple myeloma: case series and literature review.”

It is valuable to write a manuscript on an interesting topic related to Multiple Myeloma cells and macrophages interaction in disease pathogenesis.

I have some comments that I wish that it might be helpful:

- To be able to reproduce the results and to have more statistically significant power, it is highly recommended to run the same experiment on at least 6 bone marrow of patients’ myeloma samples and at least 3 bone marrow of Healthy controls’ samples. 

- Focusing on the role of UPA-UPAR axis and their role in MM disease pathogenesis via macrophages polarization, more immunological and molecular markers are favorable to address the conclusion preciously (as an example: expression of oncogenes, activation, and inhibitory genes expression).

- Figure 1: MM showed a higher subset of M1 macrophages compared to healthy controls, which is in line with your results, I think as solid proof of intracellular staining of IL-6 and HGF to these cells would support the hypothesis strongly.

- Figure 2: flow cytometric analysis needs to be reproduced because of:

o Debris in FSC/SSC figure is high showing poor staining protocol, and this may be due to lack of fixation of paraformaldehyde 1% or in appropriate washing.

o Although CD90 is a crucial marker for mesenchymal stem cells which is not necessary to be expressed on monocytes. As known as, M1 macrophages immunophenotyping is CD45+ CD14+ HLA-DR+ CD68+ CD80+ CD206- CD163-/dim and M2 macrophages immunophenotyping is CD45+ CD14+/++ HLA-DR+ CD68+ CD80- CD206+ CD163+. Could you please justify why do you involve CD90 in macrophages’ immunophenotyping?

- Line 57: a spelling mistake, correction “pro-inflammatory M1 and anti-inflammatory M2”

Author Response

Reviewer #2

Dear Authors:

Report on the review article entitled “Macrophages and urokinase plasminogen activator receptor system in multiple myeloma: case series and literature review.”

It is valuable to write a manuscript on an interesting topic related to Multiple Myeloma cells and macrophages interaction in disease pathogenesis.

I have some comments that I wish that it might be helpful:

Comment 1. To be able to reproduce the results and to have more statistically significant power, it is highly recommended to run the same experiment on at least 6 bone marrow of patients’ myeloma samples and at least 3 bone marrow of Healthy controls’ samples.

Response to Comment 1. We thank the Reviewer for this comment, and as suggested, we have increased the number of samples of six multiple myeloma patients (M/F, 4/2; mean age, 59 years old; range, 43-68 years), and four healthy controls (M/F, 3/1; mean age, 39 years old; range, 22-55 years old).

Comment 2. Focusing on the role of UPA-UPAR axis and their role in MM disease pathogenesis via macrophages polarization, more immunological and molecular markers are favorable to address the conclusion preciously (as an example: expression of oncogenes, activation, and inhibitory genes expression).

Response to Comment 2. We thank the Reviewer for this suggestion, and we have improved the discussion section as following.

On page 7, lines 210-235, the following text was added “Microenvironment plays an essential role in cancer development and progression, as also described in MM. Indeed, MM cell proliferation, survival, immunological escape, and drug resistance are deeply influenced by tumor-associated cell populations, including mesenchymal stromal cells and macrophages [72]. These cells alter tumor microenvironment with the release of tolerogenic cytokines, while also favor anti-apoptotic signaling pathway activation and drug resistance by cell-to-cell contacts with neoplastic clones. uPA/uPAR system plays an important role in several cancer development and progression, especially in favoring migration through extracellular matrix; however, several other mechanisms might be involved in MM progression thus proposing uPA/uPAR system as a candidate therapeutic target [63-64,73]. Our preliminary results showed that uPAR inhibition exerted a potent anti-inflammatory effect by almost abolishing IL-6 and DKK-1 secretion, In particular, DKK-1 glycoprotein, an inhibitory component of the Wnt pathway, promotes pathological type 2 cell-mediated inflammation, immune evasion, and induces immunosuppressive macrophages in several cancers [74-75]. In MM, DKK-1 is highly expressed by myeloma cells, and regulates the myeloma bone disease, defined as displacement of hematopoiesis and formation of osteolytic lesions, by osteoblast inhibition and osteoclast activation [76]. Moreover, DKK-1 is highly secreted in bortezomib-resistant MM, by sustaining drug resistance through CD138 downregulation, CKAP4 receptor-mediated NF-κB activation [77]. In addition, IL-6 contributes to DKK-1-mediated drug resistance, as this cytokine stimulates CKAP4 expression [77]. Therefore, simultaneous abolishment of IL-6 and DKK-1 by C6 and C37 uPAR inhibitors in our encouraging preliminary results proposes these molecules as promising targeted therapy in MM, and open new perspectives on the pathogenic role of uPAR in MM and drug resistance, as uPAR inhibitors could exert both an anti-inflammatory and a pro-immunosurveillance activity by restoring normal levels of However, our preliminary results need further validation in additional in vitro and in vivo studies.”

Comment 3. Figure 1: MM showed a higher subset of M1 macrophages compared to healthy controls, which is in line with your results, I think as solid proof of intracellular staining of IL-6 and HGF to these cells would support the hypothesis strongly.

Response to Comment 3. We really appreciate this suggestion, and we will include this experiment in further studies.

Comment 4. Figure 2: flow cytometric analysis needs to be reproduced because of: debris in FSC/SSC figure is high showing poor staining protocol, and this may be due to lack of fixation of paraformaldehyde 1% or in appropriate washing.

Response to Comment 4. We agree with the Reviewer that there were debris as shown in FSC/SSC dot plots; however, samples were not fixed as they were stained and acquired right after harvesting, and debris and double cells were removed from the analysis using linear parameters (FSC-A vs FSC-H).

Comment 6. Although CD90 is a crucial marker for mesenchymal stem cells which is not necessary to be expressed on monocytes. As known as, M1 macrophages immunophenotyping is CD45+ CD14+ HLA-DR+ CD68+ CD80+ CD206- CD163-/dim and M2 macrophages immunophenotyping is CD45+ CD14+/++ HLA-DR+ CD68+ CD80- CD206+ CD163+. Could you please justify why do you involve CD90 in macrophages’ immunophenotyping?

Response to Comment 6. We thank the Reviewer for this comment, and we completely agree with this point. However, because of technical limitations due to the availability of a two-lasers six-color flow cytometer, we optimized a five-color staining that could have allowed to simultaneously exclude mesenchymal stem cells and then identify macrophages.

Reviewer 3 Report

Thank you for sending me the research article paper “Macrophages and urokinase plasminogen activator receptor system in multiple myeloma: case series and literature review” for review in the International Journal of Molecular Science the article of Manzo al., the author role of macrophages and uPAR in multiple myeloma development, as well as discussed the therapeutic role of uPAR against multiple myeloma. However, there are important points that should be improved.

1.      Introduction part is too general. The author should explain the question needed to be solved, propose a strong hypothesis and approach to solve these questions.

2.      The introduction part is confusing. Why do authors present introductions with too many different headings, rather than present in a more scientific way of research?

3.      The Method section really needs to be explained in more detail. Author should provide information in different headings.

4.      Discussion is not enough to explain the result and prove the hypothesis.

5.      Author should include more controls in the experiments, specially in the cytokines experiment.

6.      Articles need to be completely restructured. Which type of article is it?

7.      The author should highlight the novelty and significance of this study.

Moderate editing of English language required. Most important is that the author should follow the article type format. 

Author Response

Reviewer #3

Thank you for sending me the research article paper “Macrophages and urokinase plasminogen activator receptor system in multiple myeloma: case series and literature review” for review in the International Journal of Molecular Science the article of Manzo al., the author role of macrophages and uPAR in multiple myeloma development, as well as discussed the therapeutic role of uPAR against multiple myeloma. However, there are important points that should be improved.

Comment 1. Introduction part is too general. The author should explain the question needed to be solved, propose a strong hypothesis and approach to solve these questions.

The introduction part is confusing. Why do authors present introductions with too many different headings, rather than present in a more scientific way of research?

Response to Comment 1. We thank the Reviewer for this comment, and we have completely reorganized the manuscript according to a Hypothesis paper for International Journal of Molecular Sciences journal’s format. We have shortened the introduction and added information on uPAR targeting.

Comment 2. The Method section really needs to be explained in more detail. Author should provide information in different headings.

Response to Comment 2. We thank the Reviewer for this comment, and we have completely rearranged the Materials and Methods section by also including separate paragraphs (4.1 BM-MSC culture conditions; 4.2 Flow cytometry; 4.3 Anti-uPAR treatment and cytokine detection; 4.4 Statistical analysis).

Comment 3. Discussion is not enough to explain the result and prove the hypothesis.

Response to Comment 3. We thank the Reviewer for this suggestion, and we have improved the discussion section as following.

On page 7, lines 210-235, the following text was added “Microenvironment plays an essential role in cancer development and progression, as also described in MM. Indeed, MM cell proliferation, survival, immunological escape, and drug resistance are deeply influenced by tumor-associated cell populations, including mesenchymal stromal cells and macrophages [72]. These cells alter tumor microenvironment with the release of tolerogenic cytokines, while also favor anti-apoptotic signaling pathway activation and drug resistance by cell-to-cell contacts with neoplastic clones. uPA/uPAR system plays an important role in several cancer development and progression, especially in favoring migration through extracellular matrix; however, several other mechanisms might be involved in MM progression thus proposing uPA/uPAR system as a candidate therapeutic target [63-64,73]. Our preliminary results showed that uPAR inhibition exerted a potent anti-inflammatory effect by almost abolishing IL-6 and DKK-1 secretion, In particular, DKK-1 glycoprotein, an inhibitory component of the Wnt pathway, promotes pathological type 2 cell-mediated inflammation, immune evasion, and induces immunosuppressive macrophages in several cancers [74-75]. In MM, DKK-1 is highly expressed by myeloma cells, and regulates the myeloma bone disease, defined as displacement of hematopoiesis and formation of osteolytic lesions, by osteoblast inhibition and osteoclast activation [76]. Moreover, DKK-1 is highly secreted in bortezomib-resistant MM, by sustaining drug resistance through CD138 downregulation, CKAP4 receptor-mediated NF-κB activation [77]. In addition, IL-6 contributes to DKK-1-mediated drug resistance, as this cytokine stimulates CKAP4 expression [77]. Therefore, simultaneous abolishment of IL-6 and DKK-1 by C6 and C37 uPAR inhibitors in our encouraging preliminary results proposes these molecules as promising targeted therapy in MM, and open new perspectives on the pathogenic role of uPAR in MM and drug resistance, as uPAR inhibitors could exert both an anti-inflammatory and a pro-immunosurveillance activity by restoring normal levels of However, our preliminary results need further validation in additional in vitro and in vivo studies.”

Comment 4. Author should include more controls in the experiments, specially in the cytokines experiment.

Response to Comment 4. We thank the Reviewer for this comment, and as suggested, we have increased the number of samples of six multiple myeloma patients (M/F, 4/2; mean age, 59 years old; range, 43-68 years), and four healthy controls (M/F, 3/1; mean age, 39 years old; range, 22-55 years old).

Comment 5. Articles need to be completely restructured. Which type of article is it?

Response to Comment 5. Please refer to Response to Comment 1.

Comment 6. The author should highlight the novelty and significance of this study.

Response to Comment 6. Please refer to Response to Comment 3.

Round 2

Reviewer 1 Report

Thanks to the authors for these answers.

Minor editing of English language is required

Reviewer 2 Report

Dear Authors,

Thanks for your corrections.

A strong design of experiments and good sequential steps and data analysis lead to some findings.

Good luck for your future work. 

Reviewer 3 Report

Author should include a overall graphical representation of study.

Author needs to recheck the spellings and grammar of the manuscript